# *Scoparia dulcis* L. Extract Relieved High Stocking Density-Induced Stress in Crucian Carp (*Carassius auratus*)

**DOI:** 10.3390/ani13152522

**Published:** 2023-08-04

**Authors:** Gangfu Chen, Min Wu, Huatao Li, Jing Xu, Haijing Liu, Wenhao Du, Qihui Yang, Lin Feng, Jun Jiang

**Affiliations:** 1Key Laboratory of Sichuan Province for Conservation and Utilization of Fishes Resources in the Upper Reaches of the Yangtze River, College of Life Sciences, Neijiang Normal University, Neijiang 641100, China; 2College of Fisheries, Guangdong Ocean University, Zhanjiang 524088, China; 3Animal Nutrition Institute, Sichuan Agricultural University, Chengdu 611130, China

**Keywords:** growth performance, feed intake, rollover, digestive ability, antioxidant status

## Abstract

**Simple Summary:**

High stocking density and Cu and trichlorfon exposure are typical inducers of physiological stress in fish. It is well known that the stress response may be harmful to aquatic animals, inhibiting growth, causing reproductive failure, and reducing resistance to pathogens. Hence, it is important to attenuate the detrimental effects induced by these stressors in modern farming systems. Although *Scoparia dulcis* relieved cadmium-induced oxidative stress in rats, little information is available on the effects of high stocking density and CuSO_4_ and trichlorfon exposure in fish. Therefore, this study investigates the protective effects of dietary supplementation with *Scoparia dulcis* extract on fish maintained at a high stocking density and exposed to CuSO_4_ and trichlorfon. The present study may offer a feasible way of relieving the effects induced by these stressors.

**Abstract:**

The objective of this study was to investigate the effects of *Scoparia dulcis* extract (SDE) on stress induced by high stocking density and Cu and trichlorfon exposure in crucian carp (*Carassius auratus*). The results showed that these stressors exerted detrimental effects in fish, such as inhibition of growth performance, reduced feed intake, and interruption of fish locomotion. Under high stocking density, dietary SDE supplementation increased the content of reduced glutathione (GSH) and the activities of amylase, catalase (CAT), and glutathione reductase (GR) and decreased the content of malonaldehyde (MDA) in the intestine of crucian carp. A similar trend was presented in the hepatopancreas under Cu exposure. Dietary SDE supplementation enhanced the activities of CAT, superoxide dismutase (SOD), glutathione peroxidase (GPx), lactate dehydrogenase, glutamate-oxaloacetate transaminase, and glutamate-pyruvate transaminase in the muscle of crucian carp under trichlorfon exposure. The optimum dietary SDE supplementation levels were 4.07, 4.33, and 3.95 g kg^−1^ diet based on the recovery rate of weight gain (RWG), feed intake (FI), and inhibitory rate of rollover (IR) for crucian carp under high stocking density and Cu and trichlorfon exposure, respectively. Overall, dietary supplementation with SDE may be a useful nutritional strategy for relieving these stresses in aquatic animals.

## 1. Introduction

Stocking density is a determining factor of economic return because it is directly related to fish culture productivity [1]. In practice, increasing the fish stocking density shows a positive impact on fish yield without an accompanying increase in system costs [2]. However, crowding conditions cause significant detrimental effects, indicated by reduced food consumption and body weight gain in rainbow trout [3]. The growth performance of fish is associated with the digestion and absorption of nutrients [4]. Hence, there is a probability that crowding conditions interrupt the digestion and absorption of nutrients so as to reduce fish growth performance. Furthermore, high stocking density, which serves as a common strategy and procedure in the fish farming industry, is a stressor for fish. In rainbow trout (*Oncorhynchus mykiss*), crowding conditions cause oxidative stress [3]. Therefore, these results imply that high stocking density may depress digestive ability via disruption of the oxidant-antioxidant status of fish digestive organs, which needs further investigation.

Fish health status is also negatively affected by high stocking densities [2]. Under crowded conditions, the susceptibility of fish to infections is increased. Copper sulfate (CuSO_4_) is effective and commonly applied in fish farming to control water quality and infections [5]. The hepatopancreas and intestine are important digestive and absorptive organs for fish [6]. In Jian carp (*Cyprinus carpio* var. Jian), Cu exposure causes significant detrimental effects, indicated by protein oxidation, lipid peroxidation, and alterations in the antioxidant activities of digestive and absorptive organs [7]. Therefore, it is necessary to analyze the relationship between Cu and the digestion capacity of aquatic animals.

High stocking densities increase the risk of infections induced by various pathogens, and fish are often treated with organophosphate compounds (OPCs) such as trichlorfon under this condition [8]. In practice, trichlorfon is often used excessively and it exerts toxic effects on fish mainly in the skeletal muscle [9], which plays an important role in fish body movement. Furthermore, trichlorfon exposure induces oxidative stress in the fish liver, as manifested by the decreased GSH content and activities of antioxidant enzymes as well as the increased lipid peroxidation in common carp [10]. Therefore, systematic studies are required to further examine the possible correlation between oxidative stress in fish muscle and irregular movement caused by trichlorfon.

*Scoparia dulcis*, known as sweet broomweed, has been traditionally used as a Chinese medicinal herb. It contains alkaloids, flavonoids, diterpenoids, and other biologically active substances [11]. Digestive enzymes such as amylase, trypsin, and lipase are important for nutrient digestion in fish [12]. In vitro, extracts of *S. dulcis* have shown a low inhibitory effect against amylase [13]. However, information concerning the effects of extract of *S. dulcis* (SDE) on fish kept at a high stocking density is limited. Cadmium, a kind of heavy metal, is a well-known environmental toxin [14]. A study on rats showed that dietary inclusion of *S. dulcis* relieved cadmium-induced oxidative stress injury, manifested as increased activity of superoxide dismutase (SOD) and depressed formation of malondialdehyde (MDA) in the liver [15]. The effects of SDE on the digestive and absorptive abilities of fish exposed to heavy metals such as Cu remain unclear. In the 2,2-diphenyl-1-picrylhydrazyl (DPPH^•^) reaction system, *S. dulcis* extract showed efficient radical-scavenging capacity [13]. Moreover, a study on diabetic rats showed that supplementation with *S. dulcis* extract reduced the histopathological abnormalities in the pancreas [16]. Similarly, dietary inclusion of *S. dulcis* extract relieved λ-carrageenan-induced paw edema, manifested as the increased activities of glutathione peroxidase (GPx) and superoxide dismutase (SOD) in the mouse liver and depressed formation of malondialdehyde (MDA) [17]. Notably, no other reports have focused on the relationship between SDE and antioxidant status in trichlorfon-treated fish muscle.

Therefore, the present study aimed to investigate the potential protective effects of the dietary SDE supplementation in fish maintained at a high stocking density and exposed to CuSO_4_ and trichlorfon. These effects might be related to digestive ability, the balance of antioxidant status, and bioenergetic homeostasis.

## 2. Materials and Methods

### 2.1. Chemical Reagent

Ethyl acetate (AR), acetone (AR), cyclohexane (AR), and CuSO_4_·5H_2_O (AR) were purchased from Chengdu Kelong Chemical Reagent Factory (Chengdu, China). Trichlorfon (≥90%) was provided by Shanghai Biochemical Reagent Co., Ltd. (Shanghai, China). All other chemicals used in this experiment were of analytical reagent grade (AR).

### 2.2. Preparation of SDE

*S. dulcis* was purchased from the Chengdu Pharmaceuticals Market of China (Chengdu, Sichuan, China) and identified at Neijiang Normal University, where researchers assigned the voucher samples a reference number and subsequently deposited them. The procedures for the SDE preparation were performed as previously described by our group [18]. The dry cyclohexane extract (CHE), ethyl acetate extract (EAE), acetone extract (EE), and aqueous extract (AQE) were kept in a dark place in sealed bottles and preserved at −80 °C until use.

### 2.3. Determination of Flavonoid Content

The flavonoid content of SDE was determined according to the methods of Jia et al. [19]. The flavonoid content of SDE is shown in Table 1.

### 2.4. Experimental Fish and Diets

Juvenile crucian carp (*Carassius auratus*) were purchased from a fish farm in Neijiang (Sichuan, China). Fish were maintained under lab conditions (22.0 ± 1 °C) with a natural light–dark cycle [20]. The procedures for the diet preparation were performed as previously conducted by our group [20]. The composition and nutrient content of the basal diet is given in Table 2. The basal diet contained a protein content of 34.83% and lipid content of 5.51%. The experimental diets were supplemented with AE at 0.0, 1.0, 2.0, 3.0, 4.0, 5.0, and 6.0 g kg^−1^ of diet.

### 2.5. Protection of Dietary AE against High Stocking Density-Induced Stress Assay

The procedures of the density assay were performed as previously described by our group [18]. Juvenile crucian carp (8.3 ± 0.3 g) were randomly assigned into eight groups, each of which contained 4 replicate aquariums. Group one (control) contained 15 fish per aquarium (30 cm × 30 cm × 40 cm) (normal density, 0.48 fish L^−1^), while the other groups contained 30 fish per aquarium (high density, 0.97 fish L^−1^). The basal diet was fed to the control group, and experimental diets supplemented with AE at 0.0, 1.0, 2.0, 3.0, 4.0, 5.0, and 6.0 g kg^−1^ were fed to groups two to eight for 60 days. Fish were fed six times per day. Half an hour after feeding, the uneaten feed was captured by siphoning, then dried and reweighed to calculate the feed intake (FI). In each aquarium, the water flow rate was maintained at 1.0 liters/hour. The water temperature, dissolved oxygen, and pH were 25.0 ± 1 °C, 5.0 ± 0.3 mg L^−1^, and 7.3 ± 0.3, respectively. At the beginning and end of the experiment, fish were weighed and counted for the following calculations.
Survival rate (SR) (%) = 100 × (final number/initial fish number)
Feed intake (FI) (g/fish) = feed consumed (g)/final fish number
Weight gain (WG) (g/fish) = harvest weight (g/fish) − initial weight (g/fish)
Feed efficiency (FE) (%) =100 × WG/FI
Specific growth rate (SGR) (%/day) = 100 × (ln harvest weight − ln initial weight)/experimental duration day

The sample collection procedures were performed as previously conducted by our group [4]. At the end of the trial, fish were anesthetized using benzocaine (50 mg L^−1^) and euthanized by cervical dissection, and then the intestines were immediately collected and stored at −80 °C for analysis. Protein content and lipase and amylase activities were measured. Contents of hydrogen peroxide (H_2_O_2_) and reduced glutathione (GSH), capacity of anti-superoxide anion (ASA), and activities of superoxide dismutase (SOD), catalase (CAT), and glutathione reductase (GR) were also determined.

### 2.6. Protection of Dietary AE against CuSO_4_ Exposure-Induced Stress Assay

The procedures of CuSO_4_ exposure were performed as previously described by our group [18]. A total of 420 juvenile crucian carp (4.3 ± 0.1 g) were randomly assigned into seven groups, each of which contained 4 replicate aquariums (15 fish in each aquarium). Fish were fed six times per day with the diets supplemented with AE at 0.0, 1.0, 2.0, 3.0, 4.0, 5.0, and 6.0 g kg^−1^ for 60 days, respectively. In each aquarium (30 cm × 30 cm × 40 cm), the water flow rate was maintained at 1.0 liters/hour. The water temperature, dissolved oxygen, and pH were 26.0 ± 1 °C, 5.0 ± 0.3 mg L^−1^, and 7.3 ± 0.3, respectively. After that, 30 fish from each group were randomly divided into 7 groups, each of which contained 10 fish, and they were maintained in a Cu concentration of 0.7 mg L^−1^ for four days. Additionally, 30 fish in the control group were maintained in clean water. After that, the FI and fish daily mortality in each treatment group were confirmed, and fish were anesthetized and euthanized as in the Protection of Dietary AE Against High Stocking Density-Induced Stress Assay. Then, the hepatopancreases were collected and stored at −80 °C for the analysis of amylase, lipase, and CAT activities. MDA and GSH levels were also determined.

### 2.7. Protection of Dietary AE against Trichlorfon Exposure-Induced Stress Assay

The procedures of trichlorfon exposure were performed as previously conducted by our group [18,21]. The feeding trial was the same as that in the Protection of Dietary AE Against CuSO_4_ Exposure-Induced Stress Assay.

After the feeding trial, 30 fish from each group, each of which contained 10 fish, were maintained in 2.2 mg trichlorfon L^−1^ water for four days. Another 30 fish (control group) were maintained in clean water. The rollover rate (loss of equilibrium, % of total) of each group and fish daily mortality were recorded. At the end of the trial, fish were anesthetized and euthanized, the dorsal fillets of fish were immediately collected after skinning, and they were stored at −80 °C for analysis. Content of MDA and activities of lactate dehydrogenase (LDH), glutamate-oxaloacetate transaminase (GOT), glutamate-pyruvate transaminase (GPT), SOD, GPx, and CAT were determined.

### 2.8. Biochemical Analysis

The level of H_2_O_2_ and activities of lipase, amylase, GOT, and GPT were determined according to our previous study [4,22]. Capacity of ASA, levels of MDA and GSH, and activities of LDH, SOD, CAT, GR and GPx were determined as described by Jiang et al. [7]. MDA reacts with thiobarbituric acid and can be measured at 532 nm. ASA capacity was expressed as U of activity per gram. GR activity was determined at 340 nm due to NADPH oxidation.

### 2.9. Statistical Analysis

All data (mean ± SD) were subjected to one-way or two-way ANOVA using SPSS 13.0. Duncan’s multiple range test was performed to compare treatment means. A broken-line model was used to determine the optimum dietary SDE supplementation levels under stress.

## 3. Results

### 3.1. Effects of Dietary AE on Fish Growth Performance under High Stocking Density

As shown in Table 3, fish survival rate (SR) was not influenced (100%) (*p* > 0.05), while FBW, FI, WG, FE, and SGR were significantly reduced (*p* < 0.05) under high stocking density treatment. FI, SGR, FBW, and WG were significantly enhanced with the increase in dietary AE levels up to 4.0 g kg^−1^ (*p* < 0.05), and plateaued thereafter with a further increase in dietary AE content (*p* > 0.05). A similar trend was found for FE, which improved with the increase in dietary AE levels up to 3.0 g kg^−1^. Based on broken-line analysis, the optimum dietary AE supplementation was 4.07 g kg^−1^ diet, estimated from the recovery rate of weight gain (RWG) for crucian carp under high stocking density (Figure 1).

### 3.2. Effects of Dietary AE on Fish Digestive and Absorptive Enzymatic Activities and Antioxidant Capacity under High Stocking Density

As shown in Table 4, the activities of lipase and amylase in the intestine were significantly reduced under high stocking density treatment (*p* < 0.05). The activities of lipase and amylase were significantly enhanced with the increase in dietary AE levels up to 4.0 g kg^−1^ (*p* < 0.05). The activity of SOD and ASA capacity were not influenced under high stocking density treatment (*p* > 0.05). High stocking density treatment led an increase of H_2_O_2_ level and a decrease in GSH level and CAT and GR activities in the crucian carp intestine (*p* < 0.05). The GSH content and GR activity were significantly enhanced with the increase in dietary AE levels up to 5.0 g kg^−1^ (*p* < 0.05), and plateaued thereafter with a further increase in dietary AE content (*p* > 0.05). The H_2_O_2_ content was not influenced with the increase in dietary AE levels up to 3.0 g kg^−1^ (*p* > 0.05), and it decreased with a further increase in dietary AE content. The ASA capacity and CAT activity in the intestine of fish were higher when fish were fed diets with AE at 4.0 and 5.0 g kg^−1^, respectively. The SOD activity was higher when fish were fed diets with AE at 2.0 and 3.0 g kg^−1^.

### 3.3. Effects of Dietary AE on Feed Intake of Fish under Cu Exposure 

No mortality was observed during Cu exposure. As shown in Table 5, feed intake (FI) was significantly reduced under Cu exposure (*p* < 0.05). FI was significantly enhanced with the increase in dietary AE levels up to 4.0 g kg^−1^ (*p* < 0.05), and plateaued thereafter with a further increase in dietary AE content (*p* > 0.05). Based on broken-line analysis, the optimum dietary AE supplementation was 4.33 g kg^−1^ diet, estimated from the recovery rate of feed intake (RFI) for crucian carp under Cu exposure (Figure 2).

### 3.4. Effects of Dietary AE on Digestive and Antioxidant Parameters in Fish under Cu Exposure 

As shown in Table 6, Cu exposure led to a slight increase in lipase activity in the crucian carp hepatopancreas. The activity of amylase in the hepatopancreas was significantly reduced under Cu exposure (*p* < 0.05). The activity of lipase was significantly enhanced with the increase in dietary AE levels up to 3.0 g kg^−1^ (*p* < 0.05), and plateaued thereafter with a further increase in dietary AE content. The activity of amylase in the hepatopancreas of fish fed the diet with 4.0 g kg^−1^ AE was higher than other supplementation levels. The activity of CAT and GSH content in the hepatopancreas were significantly reduced, while the level of MDA was significantly increased under Cu exposure (*p* < 0.05). The GSH content in the hepatopancreas of fish fed the diet with 3.0 g kg^−1^ AE was higher than other supplementation levels (*p* < 0.05), while the MDA content in the hepatopancreas of fish fed the diet with 4.0 g kg^−1^ AE was lower than other supplementation levels (*p* < 0.05). The CAT activity was higher when fish were fed diets with AE at 4.0 and 5.0 g kg^−1^.

### 3.5. Effects of Dietary AE on Rollover Rate of Fish under Trichlorfon-Induced Stress 

No mortality was observed during trichlorfon exposure. As shown in Table 7, the rollover rate was significantly increased under trichlorfon-induced stress (*p* < 0.05). The rollover rate was significantly reduced with the increase in dietary AE levels up to 4.0 g kg^−1^ (*p* < 0.05), and plateaued thereafter with a further increase in dietary AE content (*p* > 0.05). Based on broken-line analysis, the optimum dietary AE supplementation was 3.95 g kg^−1^ diet, estimated from the inhibitory rate of rollover (IR) for crucian carp under trichlorfon-induced stress (Figure 3).

### 3.6. Effects of Dietary AE on Metabolic Parameters and Antioxidant Status in Fish Muscle under Trichlorfon-Induced Stress 

As shown in Table 8, the activities of LDH, GPT, GOT, CAT, GPx, and SOD were significantly reduced under trichlorfon-induced stress (*p* < 0.05). However, dietary AE supplementation increased these parameters. The content of MDA was significantly increased under trichlorfon-induced stress (*p* < 0.05). The activities of GPx and SOD were significantly enhanced with the increase in dietary AE levels up to 4.0 g kg^−1^ (*p* < 0.05), and plateaued thereafter with a further increase in dietary AE content (*p* > 0.05). The content of MDA was significantly decreased with the increase in dietary AE levels up to 3.0 g kg^−1^ (*p* < 0.05), and plateaued thereafter with a further increase in dietary AE content (*p* > 0.05). The activity of GOT was significantly increased with the increase in dietary AE levels up to 4.0 g kg^−1^ (*p* < 0.05), and it decreased thereafter with a further increase in dietary AE content. Similar trends were found for the activities of LDH, GPT, and CAT.

## 4. Discussion

### 4.1. Dietary SDE Relieves the Detrimental Effects of High Stocking Density and Cu Exposure on Fish Growth Performance and Feed Intake

In practice, stocking density is a critical part of aquaculture, and the optimum stocking density is largely dependent on the fish species, fish size, and water exchange rate [23]. In modern fish farming systems, fish are maintained at high stocking densities for high production and economic return [1]. In the present study, FBW, WG, SGR, FI, and FE were decreased at a high stocking density, which indicates that fish growth was depressed by the inappropriately high stocking density. A similar observation has been reported in rainbow trout, that high stocking densities had negative effects on fish feed intake and growth performance [3]. Dietary SDE supplementation increased these parameters, which suggests that dietary SDE protects against the detrimental effects caused by the excessively high stocking density. To date, few studies have investigated the relationship between dietary SDE and growth performance under high stocking densities in fish. In rats, dietary supplementation with *S. dulcis* extract attenuated the decrease in body weight induced by alloxan [24]. In the present study, based on broken-line analysis, the optimum dietary SDE supplementation was 4.07 g kg^−1^ diet estimated from the RWG for crucian carp under high stocking density.

In the fish farming industry, water quality and infections are commonly controlled by copper sulfate [5]. On the one hand, Cu is an essential metal for fish. On the other hand, as a heavy metal, it has an adverse impact on the aquatic ecosystem and humans because of its accumulation in the aquatic environment [25]. In the present study, copper caused significant detrimental effects on crucian carp, as indicated by the reduced feed intake. Consistent with our present results, a study on Indian major carp (*Cirrhinus mrigala* Hamilton) showed that feed intake in fish was significantly reduced and accompanied with morphological changes, such as being less active, in all Cu treatment groups [26]. Additionally, copper increased the mortality of crucian carp [18]. Similar observations have been reported in *Cirrhinus mrigala* [26] and Caspian Sea kutum (*Rutilus frisii kutum*) [25]. In the present study, FI was significantly decreased by CuSO_4_ exposure and dietary SDE supplementation increased these parameters, indicating that dietary SDE alleviated the detrimental effect caused by Cu exposure. In the present study, based on broken-line analysis, the optimum dietary SDE supplementation was 4.33 g kg^−1^ diet estimated from the RFI for crucian carp under Cu-induced stress.

### 4.2. Dietary SDE Relieves the Detrimental Effects of High Stocking Density and Cu Exposure on Fish Digestive Ability

Fish growth performance is associated with their nutrient digestion and absorption capacities, which depend on the activities of digestive enzymes and brush-border membrane enzymes [4]. Numerous digestive enzymes in fish are synthesized in the exocrine pancreas and then secreted into the intestinal tract [27]. Examples of such enzymes include trypsin, lipase, chymotrypsin, and amylase [12]. In this study, the activities of lipase and amylase in the fish intestine were decreased at a high stocking density, which indicates that the digestive ability of fish was depressed by the excessively high stocking density. Similar observations have been reported in tilapia (*Oreochromis niloticus*) [28], Liao river shrimp (*Palaemonetes sinensis*) [29], and largemouth bass (*Micropterus salmoides*) [30]. The decreased activities of fish digestive enzymes might have been due to the disrupted endocrine system and elevated cortisol levels induced by the high stocking densities [30]. In rainbow trout, stocking at high density decreased the expression of genes related to the function and integrity of the intestinal epithelium [31]. Our present study showed that amylase activity in the hepatopancreas was decreased under CuSO_4_ exposure, which suggests that the digestive ability of crucian carp was depressed under Cu exposure. A similar observation has been reported in juvenile Epinephelus coioides that Cu exposure hindered the activities of digestive enzymes, including protease, lipase, and amylase, in the liver and intestine [32]. The decreased activities of fish digestive enzymes might have been due to the deposited metal granules in the fish liver provoking hepatic DNA damage [33]. In this study, dietary SDE supplementation increased the activities of amylase and lipase, which indicates that dietary SDE protects against the detrimental effect on fish digestive ability caused by the excessively high stocking density and Cu exposure. However, to the best of our knowledge, our study is the first to examine whether SDE can relieve the detrimental effects on the digestive ability of fish under high stocking density- and Cu-induced stress.

### 4.3. Dietary SDE Relieves the Detrimental Effects of High Stocking Density and Cu Exposure on Antioxidant Status in Fish Digestive Organs

The foundation of digestive ability is the structural and functional integrity of fish digestive organs [4]. In aquatic animals, the structure and function of tissues and organs are largely dependent on their antioxidant status [34]. Reactive oxygen species (ROS) are generated by aerobic metabolism and exogenous sources and induce oxidative damage to lipids and proteins [35]. Lipid peroxidation, with MDA as a sensitive marker, takes place as a result of the oxidative deterioration of polyunsaturated fatty acids (PUFAs), which are widely distributed in fish bodies [34]. In the present study, the MDA content was significantly higher in the hepatopancreas of crucian carp under Cu exposure, which indicates that Cu-induced stress caused lipid peroxidation in the digestive organ. Consistent with our present results, a study on rainbow trout (*Oncorhynchus mykiss*) showed that Cu-induced stress negatively affected the liver MDA content [36]. In the present study, the MDA content was significantly increased under Cu exposure and decreased after dietary SDE supplementation, which suggests that dietary SDE diminishes the negative influence of the Cu-induced stress on lipid peroxidation in fish digestive organs. Similarly, dietary inclusion of *S. dulcis* extract relieved the streptozotocin-induced oxidative stress in the liver and pancreas of diabetic rats, manifested as the depressed formation of malondialdehyde (MDA) [16,37]. Moreover, the study on diabetic rats showed that supplementation with *S. dulcis* extract reduced histopathological abnormalities in the pancreas [16]. The beneficial effect of *S. dulcis* extract on the inhibition of lipid peroxidation may be related to flavonoids. Lipid peroxidation, an enzymatical process, is associated with enzymes such as xanthine oxidase and lipoxygenase [38]. In addition, flavonoids inhibit the lipid peroxidation mediated by xanthine oxidase and lipoxygenase [39,40].

The increased generation of ROS, namely hydrogen peroxide (H_2_O_2_), superoxide radicals (O_2_^•−^), and hydroxyl radicals (^•^OH), can induce lipid peroxidation [7]. In the present study, the level of hydrogen peroxide (H_2_O_2_) in the fish intestine was increased at a high stocking density. Dietary SDE supplementation decreased the hydrogen peroxide (H_2_O_2_) level and increased the ASA capacity (indicated O_2_^•−^-scavenging ability), which indicates that dietary SDE depressed the ROS generation caused by the excessively high stocking density. Similarly, treatment with extract of *S. dulcis* decreased the level of hydroperoxide in the liver of diabetic rats [37]. In addition, *S. dulcis* extract displayed O_2_^•−^-scavenging ability and mitigated the generation of ^•^OH in vitro [41,42]. In the DPPH^•^ reaction system, *S. dulcis* extract showed efficient radical-scavenging capacity [13]. Furthermore, *S. dulcis* extract may play an important role in preventing the generation of ROS rather than neutralizing the ROS that are already produced [43]. In the DPPH^•^ reaction system, the higher radical-scavenging capacity was directly related to the higher flavonoid content in *S. dulcis* extract [13]. Therefore, flavonoids might have a favorable effect on inhibiting ROS generation in fish digestive organs, which needs further investigation.

The antioxidant defense system in fish, consisting of antioxidant enzymes and non-enzymatic antioxidants, has developed to avoid or repair the damage caused by ROS to fish tissues and organisms [44]. In the antioxidant enzyme system, SOD is the first enzyme to respond against O_2_^•−^, catalyzing it to H_2_O_2_ and the dioxygen molecule [34]. Then, H_2_O_2_ can be detoxified by CAT and GPx [30]. The glutathione antioxidant system, an effective cellular defense against oxidative stress, consists of glutathione S-transferase (GST), glutathione reductase (GR), and a low molecular compound, namely GSH [44]. At the expense of NADPH, GR catalyzes the oxidized glutathione (GSSG) to GSH [45]. Our present study showed that the activities of CAT and GR were decreased in the intestine at a high stocking density, which indicates that the inappropriately high stocking density had adverse effects on the antioxidant enzyme defense system in fish digestive organs. A similar result has been observed in largemouth bass (*Micropterus salmoides*) [30]. Meanwhile, the present study showed that the activity of CAT was also decreased in the crucian carp hepatopancreas under Cu-induced stress. Similarly, Cu exposure decreased the activities of CAT and GPx in the rainbow trout liver [36]. GSH, a major non-enzymatic antioxidant, is important for scavenging intracellular ROS [46]. Our present study showed that the GSH content was significantly decreased in the crucian carp hepatopancreas and intestine under Cu exposure and high stocking density, which suggests that Cu- and high stocking density-induced stress had adverse effects on the non-enzymatic antioxidant defense system in fish digestive organs. Similarly, a study on rainbow trout showed that Cu exposure decreased the GSH concentration in the fish liver [36]. In this study, dietary SDE supplementation increased the SOD, CAT, and GR activities and GSH content in the digestive organs of fish under high stocking density- and Cu-induced stress, indicating that dietary SDE alleviated the detrimental effect on the fish antioxidant defense system caused by these stressors. Similar observations have been reported that the dietary inclusion of *S. dulcis* extract relieved the streptozotocin-induced oxidative stress injury, manifested as increased activities of SOD, GPx, CAT, and GR and GSH content in the rat liver or pancreas [16,37,47]. Our results suggest that dietary SDE enhances fish digestive ability via regulating the oxidant-antioxidant status of fish digestive organs under high stocking density- and Cu-induced stress. The beneficial effect of *S. dulcis* extract on the fish antioxidant defense system may be related to flavonoids. In the liver of the dark sleeper (*Odontobutis potamophila*), the activities of SOD, GST and GPx were increased by the dietary inclusion of quercetin, a kind of flavonoid compound [48]. Acacetin, an O-methylated flavone, protected against CuSO_4_-induced stress, as manifested by the reduced ROS and lipid peroxidation levels and the upregulation of antioxidant genes such as GPx and GR [49]. Therefore, flavonoids might have a favorable effect on reducing oxidative stress in fish digestive organs, which needs further investigation.

### 4.4. Dietary SDE Relieves the Detrimental Effects of Trichlorfon Exposure on Fish Muscle Function and Bioenergetic Homeostasis

Trichlorfon causes fish balance loss by inhibiting acetylcholinesterase (AChE) activity at the neuromuscular junctions of fish muscles [9]. In the present study, trichlorfon induced significant detrimental effects, as indicated by the crucian carp rollover rate. The frequency of this phenomenon was relieved by dietary SDE supplementation, which indicates that dietary SDE alleviated the detrimental effect caused by trichlorfon. In the present study, the optimum dietary SDE supplementation, estimated from IR using broken-line analysis, was 3.95 g kg^−1^ diet for crucian carp.

Fish movements are largely dependent on the energy metabolism of swimming muscles [50]. LDH, a general biomarker of stress, is responsible for the reversible transformation of pyruvate to lactate. It is a crucial enzyme for energy production in the anaerobic pathway [51]. In the present study, the LDH activity was significantly lower in crucian carp muscle exposed to trichlorfon, contributing to an energy imbalance. A similar observation has been reported in silver catfish (*Rhamdia quelen*) muscle cytosol and mitochondria that exposure to trichlorfon altered the bioenergetic homeostasis by inhibiting the activity of creatine kinase (an important enzyme for energy homeostasis) and activating complexes II-III and IV of the respiratory chain (a crucial pathway for ATP production) [52].

GOT and GPT are two vital enzymes involved in amino acid metabolism that facilitate the utilization of amino acids as sources of energy by deamination [4]. In the present study, the GOT and GPT activities were also significantly lower in crucian carp muscle exposed to trichlorfon, contributing to an energy imbalance. Generally, fish utilize protein for energy supply under normal conditions [53]. Under stressful conditions, such as exposure to pesticides, lipids are used as an alternative energy source in fish muscle, corroborating the alteration in the fatty acid profile of silver catfish in the presence of trichlorfon [52]. In this study, the activities of GPT, LDH, and GOT were decreased in crucian carp muscle under trichlorfon exposure and increased after dietary SDE supplementation, which indicates that dietary SDE protects against the inhibitory effect of trichlorfon on fish bioenergetic homeostasis. This may partly explain the trichlorfon-induced rollover of crucian carp. However, the mechanism by which SDE relieved the trichlorfon-induced muscle dysfunction in fish needs further exploration.

### 4.5. Dietary SDE Relieves the Detrimental Effect of Trichlorfon Exposure on Antioxidant Status in Fish Muscle

Trichlorfon exposure increases ROS in fish, resulting in protein oxidation and lipid peroxidation [8]. Our present study showed that the MDA content in crucian carp muscle was significantly higher under trichlorfon exposure, contributing to lipid peroxidation. A similar observation has been reported that trichlorfon induced lipid oxidative damage, which impaired gill function and cell structure in common carp (*Cyprinus carpio* L.) [54]. In this study, dietary SDE supplementation decreased the values of these parameters, indicating that it weakened the trichlorfon-induced damage. Our results align with the report on diabetic rats in which dietary inclusion of *S. dulcis* extract relieved streptozotocin-induced lipid peroxidation in the brain, manifested as decreased thiobarbituric acid reactive substance (TBARS) and hydroperoxide levels [55].

The antioxidant defense system in fish, consisting of enzymatic and non-enzymatic antioxidants, is crucial for the protection against oxidative stress in biological defense systems [44]. In biological systems, O_2_^•−^ plays a fundamental role in oxidative stress and damage, resulting in lipid peroxidation [56]. In the antioxidant enzyme system, SOD is the first enzyme to respond against O_2_^•−^, catalyzing it to H_2_O_2_ and the dioxygen molecule [34]. Then, H_2_O_2_ can be detoxified by CAT and GPx [30]. SOD, GPx, and CAT are three key enzymes in the response against O_2_^•−^ and they are categorized as first line defense antioxidants [56]. In this study, SOD, CAT, and GPx activities were significantly lower in crucian carp muscle exposed to trichlorfon, which indicates that the fish enzymatic antioxidant defense system was disturbed by trichlorfon exposure, resulting in lipid peroxidation in fish muscle. A similar observation has been reported that trichlorfon exposure decreased the CAT activity in the common carp (*Cyprinus carpio*) gill and liver [54]. The glutathione (GSH) antioxidant system, comprising GPx, GR, and GST, is an effective cellular defense against oxidative stress [44]. In order to eliminate H_2_O_2_, GSH acts as a co-substrate with GPx [57]. Deltamethrin, a synthetic pyrethroid pesticide, decreased the GSH level in the muscle of a freshwater teleost, Channa punctata [58]. In the present study, dietary SDE supplementation increased these parameters, which indicates that dietary SDE protects against the detrimental effect of the trichlorfon-induced imbalance between oxidants and antioxidants. The beneficial effect of *S. dulcis* extract on the antioxidant defense system in fish muscle may be related to flavonoids. In the freshwater teleost Channa punctata, the deltamethrin-induced inhibition of GSH was alleviated by treatment with quercetin, a kind of flavonoid compound [58]. This preventive effect may have been due to its antioxidant properties. It is hypothesized that flavonoids may enhance the antioxidant defense system in fish muscle. However, to the best of our knowledge, our study is the first to demonstrate that SDE relieves the detrimental effect of imbalance between oxidants and antioxidants in fish muscle under trichlorfon exposure.

## 5. Conclusions

Under the inappropriately high stocking density, fish growth was depressed, the digestive capacity was degraded, and the antioxidant status was interrupted. These parameters were improved after dietary SDE supplementation, indicating that it relieved the detrimental effect of high stocking density on fish growth performance. Cu exposure reduced fish feed intake, while SDE supplementation mitigated the disruptive effects of Cu-induced stress on the digestive capacity as well as the antioxidant status of fish. Based on broken-line analysis, the optimum dietary SDE supplementation levels were 4.07 and 4.33 g kg^−1^ diet, estimated from the RWG and RFI for crucian carp under high stocking density- and Cu-induced stress, respectively. Additionally, trichlorfon exposure induced detrimental effects, as indicated by the rollover rate of crucian carp. Specifically, the bioenergetic homeostasis and antioxidant status in fish muscle were interrupted by trichlorfon. The negative influence was mitigated after dietary SDE supplementation at the optimum content of 3.95 g kg^−1^ diet, estimated from the IR. Dietary SDE supplementation offers a feasible way of relieving the stress induced by inappropriately high stocking density, Cu exposure, and trichlorfon exposure. However, the specific mechanisms need to be further explored.

## Figures and Tables

**Figure 1 animals-13-02522-f001:**
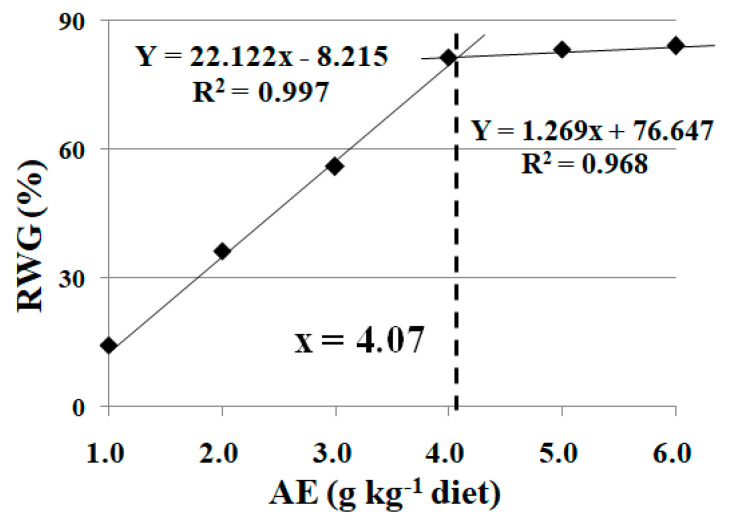
Broken-line analysis of recovery rate of weight gain (RWG) for crucian carp at high stocking density fed diets containing different levels of acetone extract of *S. dulcis* (AE) for 60 days. RWG = 100 × (E − Y)/(K − Y) (in Table 3). Values are mean ± SD of 4 replicates.

**Figure 2 animals-13-02522-f002:**
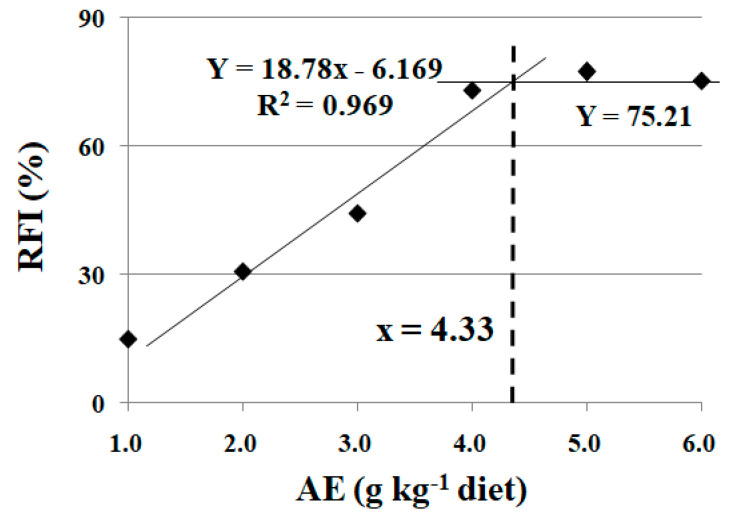
Broken-line analysis of recovery rate of feed intake (RFI) for crucian carp fed diets containing different levels of acetone extract of *S. dulcis* (AE) for 60 days, followed by Cu exposure for 4 days. RFI = 100 × (E − Y)/(K − Y) (in Table 5). Values are mean ± SD of three replicates, with 10 fish in each replicate.

**Figure 3 animals-13-02522-f003:**
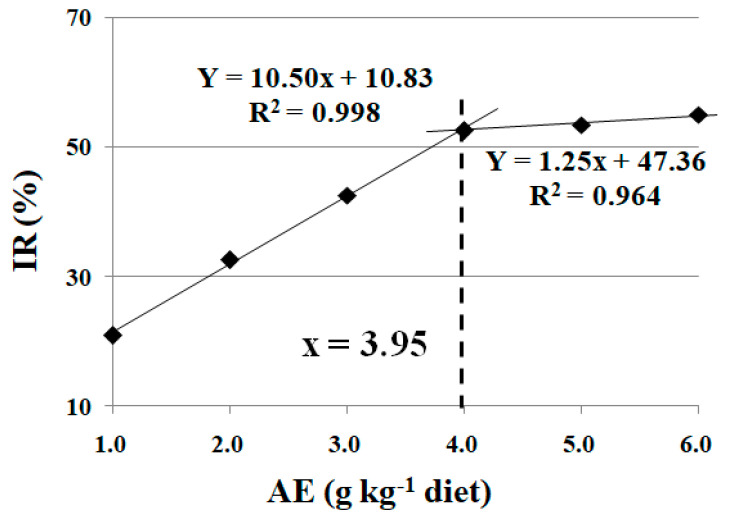
Broken-line analysis of inhibitory rate of rollover (IR) for crucian carp fed diets containing different levels of acetone extract of *S. dulcis* (AE) for 60 days, followed by trichlorfon exposure for 4 days. IR = Y − E (in Table 7). Values are mean ± SD of three replicates, with 10 fish in each replicate.

**Table 1 animals-13-02522-t001:** Flavonoid contents of cyclohexane extract (CHE), ethyl acetate extract (EAE), acetone extract (AE), and aqueous extract (AQE) of *S. dulcis*.

Extracts	Flavonoids (mg g^−1^ Dry Extract)
CHE	30.22 ± 1.68 ^b^
EAE	38.66 ± 1.91 ^c^
AE	67.67 ± 3.36 ^d^
AQE	21.40 ± 1.27 ^a^

Values are mean ± S.D. of 3 replicates. Values in the same column with different superscripts are significantly different (*p* < 0.05).

**Table 2 animals-13-02522-t002:** Composition and nutrient content of the basal diet.

Ingredients	%	Proximate Analysis ^3^	%
Fish meal	25.0	Dry matter	93.16
Soybean meal	32.0	Crude protein	34.83
Wheat flour	37.0	Crude lipid	5.51
DL-methionine	0.70	Crude Ash	5.87
Fish oil	1.50		
Sunflower oil	1.80		
Vitamin mixture ^1^	1.00		
Mineral mixture ^2^	1.00		

^1^ Per kg of vitamin mix: retinyl acetate (500,000 IU g^−1^), 0.80 g; cholecalciferol (500,000 IU g^−1^), 0.48 g; DL-α-tocopherol acetate (50%), 20.00 g; menadione (23%), 0.43 g; thiamin nitrate (90%), 0.11 g; riboflavine (80%), 0.63 g; pyridoxine HCl (81%), 0.92 g; cyanocobalamin (1%), 0.10 g; ascorhyl acetate (93%), 7.16 g; D-calcium pantothenate (90%), 2.73 g; niacin (99%), 2.82 g; D-biotin (2%), 5.00 g; meso-inositol (99%), 52.33 g; folic acid (96%), 0.52 g. ^2^ Per kg of mineral mix: FeSO_4_·7H_2_O (20% Fe), 69.70 g; CuSO_4_·5H_2_O (25% Cu), 1.20 g; ZnSO_4_·7H_2_O (23% Zn), 21.64 g; MnSO_4_·H_2_O (32% Mn), 4.09 g; Na_2_SeO_3_·5H_2_O (1% Se), 2.50 g; KI (4% I), 2.90 g; CaCO_3_, 897.98 g. ^3^ Proximate analysis was performed according to the procedures described by Chen et al. [4].

**Table 3 animals-13-02522-t003:** Initial body weight (IBW), final body weight (FBW), weight gain (WG), specific growth rate (SGR), feed intake (FI), feed efficiency (FE), and survival rate (SR) of crucian carp at high stocking density fed diets containing different levels of acetone extract of *S. dulcis* (AE) for 60 days.

Densities (Fish L^−1^) + AE (g kg^−1^ Diet)	IBW (g Fish^−1^)	FBW (g Fish^−1^)	WG (g Fish^−1^)	SGR (% d^−1^)	FI (g Fish^−1^)	FE (%)	SR (%)
0.48 + 0 (K)	8.24 ± 0.32 ^a^	30.96 ±1.38 ^d^	22.72 ± 1.30 ^d^	2.21 ± 0.14 ^c^	32.14 ± 1.21 ^d^	70.66 ± 4.22 ^d^	100.00 ± 0.00 ^a^
0.97 + 0 (Y)	8.25 ± 0.35 ^a^	19.74 ± 1.19 ^a^	11.49 ± 0.81 ^a^	1.45 ± 0.1 ^a^	26.98 ± 1.56 ^a^	42.80 ± 1.45 ^a^	100.00 ± 0.00 ^a^
0.97 + 1 (E_1_)	8.35 ± 0.36 ^a^	21.45 ± 0.97 ^a^	13.11 ± 0.58 ^a^	1.57 ± 0.08 ^a^	27.86 ± 1.37 ^ab^	47.17 ± 2.84 ^ab^	100.00 ± 0.00 ^a^
0.97 + 2 (E_2_)	8.34 ± 0.31 ^a^	23.92 ± 1.52 ^b^	15.58 ± 1.03 ^b^	1.75 ± 0.15 ^b^	28.98 ± 1.30 ^ab^	53.82 ± 4.13 ^bc^	100.00 ± 0.00 ^a^
0.97 + 3 (E_3_)	8.30 ± 0.41 ^a^	26.08 ± 1.29 ^c^	17.78 ± 0.99 ^c^	1.91 ± 0.11 ^b^	29.67 ± 1.50 ^bc^	60.08 ± 4.12 ^cd^	100.00 ± 0.00 ^a^
0.97 + 4 (E_4_)	8.20 ± 0.33 ^a^	28.85 ± 1.10 ^d^	20.65 ± 1.38 ^d^	2.1 ± 0.12 ^c^	30.90 ± 1.26 ^cd^	66.89 ± 4.72 ^d^	100.00 ± 0.00 ^a^
0.97 + 5 (E_5_)	8.28 ± 0.40 ^a^	29.12 ± 1.53 ^d^	20.84 ± 1.09 ^d^	2.1 ± 0.14 ^c^	31.12 ± 1.38 ^cd^	67.12 ± 2.93 ^d^	100.00 ± 0.00 ^a^
0.97 + 6 (E_6_)	8.16 ± 0.18 ^a^	29.10 ± 1.42 ^d^	20.94 ± 1.37 ^d^	2.12 ± 0.08 ^c^	31.13 ± 1.48 ^cd^	67.44 ± 4.02 ^d^	100.00 ± 0.00 ^a^

Values are mean ± SD of 4 replicates. Values in the same column with the different superscripts are significantly different (*p* < 0.05).

**Table 4 animals-13-02522-t004:** The anti-superoxide anion (ASA) capacity, activities of amylase, lipase, superoxide dismutase (SOD), catalase (CAT), and glutathione reductase (GR), and contents of hydrogen peroxide (H_2_O_2_) and reduced glutathione (GSH) in the intestine of crucian carp at high stocking density fed diets containing different levels of acetone extract of *S. dulcis* (AE) for 60 days.

Densities (Fish L^−1^) + AE (g kg^−1^ Diet)	Amylase (U mg^−1^ Protein)	Lipase (U mg^−1^ Protein)	ASA (U g^−1^ Protein)	H_2_O_2_ (mmol g^−1^ Protein)	SOD (U mg^−1^ rotein)	CAT (U mg^−1^ Protein)	GSH (mg g^−1^ Protein)	GR (U g^−1^ Protein)
0.48 + 0 (K)	1.43 ± 0.07 ^c^	46.22 ± 2.58 ^d^	35.91 ± 2.87 ^a^	16.24 ± 0.86 ^a^	97.64 ± 7 ^a^	27.09 ± 1.7 ^b^	14.24 ± 0.92 ^c^	14.37 ± 0.88 ^bc^
0.97 + 0 (Y)	0.91 ± 0.06 ^a^	30.53 ± 2.64 ^a^	37.54 ± 1.62 ^a^	29.62 ± 2.01 ^c^	100.53 ± 3.4 ^a^	20.41 ± 1.13 ^a^	6.42 ± 0.43 ^a^	10.23 ± 0.9 ^a^
0.97 + 1 (E_1_)	0.94 ± 0.07 ^a^	33.27 ± 2.62 ^a^	37.49 ± 1.84 ^a^	28.3 ± 1.56 ^c^	99.85 ± 7.38 ^a^	20.5 ± 0.92 ^a^	6.36 ± 0.26 ^a^	12.67 ± 0.9 ^b^
0.97 + 2 (E_2_)	1.02 ± 0.07 ^a^	38.26 ± 2.65 ^b^	46.14 ± 2.71 ^b^	26.88 ± 1.58 ^c^	132.88 ± 9.99 ^c^	25.21 ± 1.43 ^b^	6.85 ± 0.43 ^ab^	12.82 ± 0.91 ^b^
0.97 + 3 (E_3_)	1.29 ± 0.07 ^b^	38.63 ± 2.68 ^bc^	49.17 ± 3.02 ^bc^	27.02 ± 1.16 ^c^	134.44 ± 10.92 ^c^	26.62 ± 1.55 ^b^	8.4 ± 0.53 ^b^	14.24 ± 0.92 ^bc^
0.97 + 4 (E_4_)	1.4 ± 0.06 ^bc^	43.37 ± 2.68 ^cd^	51.87 ± 3.03 ^c^	19.84 ± 1.1 ^b^	123.56 ± 6.55 ^bc^	27.27 ± 1.69 ^b^	15.01 ± 0.7 ^c^	14.27 ± 0.92 ^bc^
0.97 + 5 (E_5_)	1.45 ± 0.05 ^c^	43.54 ± 2.6 ^cd^	47.94 ± 2.97 ^bc^	15.78 ± 0.59 ^a^	125.22 ± 7.48 ^bc^	31.24 ± 2.03 ^c^	19.49 ± 1.19 ^d^	15.72 ± 0.89 ^c^
0.97 + 6 (E_6_)	1.35 ± 0.09 ^bc^	40.56 ± 2.7 ^bc^	45.7 ± 1.66 ^b^	14.72 ± 1.06 ^a^	112.61 ± 6.03 ^ab^	28.64 ± 1.9 ^bc^	18.61 ± 1.09 ^d^	15.03 ± 0.92 ^c^

Values are mean ± SD of 4 replicates, with 5 fish in each replicate. Values with the different superscripts in the same column are significantly different (*p* < 0.05).

**Table 5 animals-13-02522-t005:** Feed intake (FI) of crucian carp fed diets containing different levels of acetone extract of *S. dulcis* (AE) for 60 days, followed by Cu exposure for 4 days.

AE (g kg^−1^ Diet) + Cu (mg L^−1^)	FI (% of Body Weight)
0 + 0.0 (K)	4.25 ± 0.09 ^f^
0 + 0.7 (Y)	0.07 ± 0.00 ^a^
1 + 0.7 (E_1_)	0.70 ± 0.01 ^b^
2 + 0.7 (E_2_)	1.35 ± 0.12 ^c^
3 + 0.7 (E_3_)	1.93 ± 0.14 ^d^
4 + 0.7 (E_4_)	3.12 ± 0.17 ^e^
5 + 0.7 (E_5_)	3.30 ± 0.19 ^e^
6 + 0.7 (E_6_)	3.22 ± 0.26 ^e^

Values are mean ± SD of three replicates, with 10 fish in each replicate. Values in the same column with different superscripts are significantly different (*p* < 0.05).

**Table 6 animals-13-02522-t006:** The activities of amylase, lipase, and catalase (CAT) and the contents of malondialdehyde (MDA) and reduced glutathione (GSH) in the hepatopancreas of crucian carp fed diets containing different levels of acetone extract of *S. dulcis* (AE) for 60 days, followed by exposure to Cu for 4 days.

AE (g kg^−1^) + Cu (mg L^−1^)	Amylase (U mg^−1^ Protein)	Lipase (U g^−1^ Protein)	MDA (nmol mg^−1^ Protein)	CAT (U mg^−1^ Protein)	GSH (mg g^−1^ Protein)
0 + 0.0 (K)	1.02 ± 0.03 ^b^	26.95 ± 1.47 ^a^	12.28 ± 0.71 ^a^	36.33 ± 1.94 ^d^	13.87 ± 0.79 ^e^
0 + 0.7 (Y)	0.78 ± 0.02 ^a^	28.13 ± 1.53 ^ab^	21.81 ± 1.18 ^d^	27.66 ± 1.84 ^a^	9.2 ± 0.42 ^a^
1 + 0.7 (E_1_)	0.98 ± 0.02 ^b^	29.19 ± 1.47 ^ab^	19.11 ± 0.85 ^c^	28.71 ± 1.2 ^ab^	9.95 ± 0.61 ^ab^
2 + 0.7 (E_2_)	1.04 ± 0.05 ^b^	33.64 ± 1.53 ^bc^	19.28 ± 1.23 ^c^	30.38 ± 0.91 ^abc^	11.48 ± 0.77 ^cd^
3 + 0.7 (E_3_)	1.03 ± 0.03 ^b^	38.24 ± 3.87 ^c^	15.80 ± 1.00 ^b^	31.46 ± 1.03 ^bc^	12.17 ± 0.40 ^d^
4 + 0.7 (E_4_)	1.11 ± 0.04 ^c^	36.9 ± 1.53 ^c^	13.28 ± 0.87 ^a^	33.16 ± 1.46 ^c^	11.32 ± 0.60 ^cd^
5 + 0.7 (E_5_)	1.05 ± 0.03 ^bc^	37.75 ± 4.04 ^c^	13.98 ± 0.43 ^ab^	32.87 ± 1.5 ^c^	10.97 ± 0.48 ^bcd^
6 + 0.7 (E_6_)	1.01 ± 0.03 ^b^	38.31 ± 4.52 ^c^	14.21 ± 0.52 ^ab^	31.94 ± 1.37 ^bc^	10.71 ± 0.41 ^bc^

Values are mean ± SD of 3 replicates, with 5 fish in each replicate. Values with the different superscripts in the same column are significantly different (*p* < 0.05).

**Table 7 animals-13-02522-t007:** Rollover rate of crucian carp fed diets containing different levels of acetone extract of *S. dulcis* (AE) for 60 days, followed by trichlorfon exposure for 4 days.

AE (g kg^−1^ Diet) + Trichlorfon (mg L^−1^)	Rollover (% of Total)
0 + 0.0 (K)	0.00 ± 0.00 ^a^
0 + 2.2 (Y)	100.00 ± 0.00 ^f^
1 + 2.2 (E_1_)	79.17 ± 2.89 ^e^
2 + 2.2 (E_2_)	67.50 ± 4.33 ^d^
3 + 2.2 (E_3_)	57.50 ± 2.50 ^c^
4 + 2.2 (E_4_)	47.50 ± 2.50 ^b^
5 + 2.2 (E_5_)	46.67 ± 2.89 ^b^
6 + 2.2 (E_6_)	45.00 ± 2.50 ^b^

Values are mean ± SD of three replicates, with 10 fish in each replicate. Values in the same column with different superscripts are significantly different (*p* < 0.05).

**Table 8 animals-13-02522-t008:** The activities of lactate dehydrogenase (LDH), glutamate-oxaloacetate transaminase (GOT), glutamate-pyruvate transaminase (GPT), superoxide dismutase (SOD), catalase (CAT), and glutathione peroxidase (GPx) and the content of malondialdehyde (MDA) in the muscle of crucian carp fed diets containing different levels of acetone extract of *S. dulcis* (AE) for 60 days, followed by exposure to trichlorfon for 4 days.

AE (g kg^−1^ Diet) + Trichlorfon (mg L^−1^)	LDH (U g^−1^ Protein)	GOT (U g^−1^ Protein)	GPT (U g^−1^ Protein)	MDA (nmol mg^−1^ Protein)	SOD (U mg^−1^ Protein)	CAT (U mg^−1^ Protein)	GPx (U mg^−1^ Protein)
0 + 0.0 (K)	292.2 ± 24.35 ^d^	14.94 ± 0.78 ^c^	16.38 ± 1.02 ^d^	3.04 ± 0.14 ^a^	152.36 ± 4.92 ^cd^	25.05 ± 0.87 ^d^	381.88 ± 31.25 ^c^
0 + 2.2 (Y)	207.4 ± 14.37 ^a^	10.21 ± 0.81 ^a^	11.44 ± 0.73 ^a^	4.71 ± 0.23 ^d^	122.17 ± 5.8 ^a^	14.08 ± 1.17 ^a^	236.98 ± 12.07 ^a^
1 + 2.2 (E_1_)	207.92 ± 13.85 ^a^	12.19 ± 1.04 ^b^	12.69 ± 0.83 ^ab^	4.64 ± 0.15 ^d^	129.36 ± 4.61 ^ab^	16.04 ± 1.13 ^a^	255.31 ± 20.16 ^a^
2 + 2.2 (E_2_)	237.35 ± 14.18 ^ab^	13.73 ± 0.76 ^bc^	14.73 ± 0.79 ^cd^	4.16 ± 0.16 ^c^	138.02 ± 9.24 ^bc^	22.23 ± 1.75 ^bc^	316.32 ± 11.91 ^b^
3 + 2.2 (E_3_)	231.91 ± 13.85 ^ab^	13.42 ± 0.85 ^bc^	14.66 ± 1.18 ^cd^	3.7 ± 0.19 ^b^	147.67 ± 12.19 ^cd^	21.97 ± 1.86 ^bc^	313.54 ± 16.91 ^b^
4 + 2.2 (E_4_)	269.13 ± 14.57 ^cd^	16.78 ± 1.24 ^d^	15.53 ± 0.83 ^cd^	3.71 ± 0.21 ^b^	163.65 ± 8.39 ^d^	23.36 ± 0.78 ^bcd^	360.37 ± 24.48 ^c^
5 + 2.2 (E_5_)	273.24 ± 13.92 ^cd^	12.39 ± 1.22 ^b^	15.18 ± 0.52 ^cd^	3.49 ± 0.15 ^b^	158.82 ± 6.96 ^d^	23.56 ± 1.14 ^cd^	369.1 ± 30.45 ^c^
6 + 2.2 (E_6_)	256.3 ± 14.32 ^bc^	12.75 ± 1.17 ^b^	14.25 ± 1.16 ^bc^	3.52 ± 0.2 ^b^	158.35 ± 11.56 ^d^	20.92 ± 1.59 ^b^	356.58 ± 21.22 ^c^

Values are mean ± SD of 3 replicates, with 5 fish in each replicate. Values with the different superscripts in the same column are significantly different (*p* < 0.05).

## Data Availability

The data used to generate the results in this manuscript can be made available if requested from the corresponding author.

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
