# Peer review of "Scoparia dulcis* L. Extract Relieved High Stocking Density-Induced Stress in Crucian Carp (*Carassius auratus*)"

_animals, 2023, doi:10.3390/ani13152522_

Round 1

Reviewer 1 Report

Overall, the manuscript is interesting, the experiment is well designed, the methods are adequate and well developed. Only minor issues /suggestions / recommendations:  

 Abstract:

The abbreviation for Scoparia dulcis extract should be 'SDE' instead of 'ESD'.

The abstract as written emphasizes the negative impact of the stressors. I suggest that the authors give more emphasis to the beneficial effects of dietary ESD.

Introduction:

In paragraphs 40-51, the authors could also include a statement suggesting that the degradation of growth and feed utilization from overcrowding may be triggered by a rising demand for energy to activate physiological functions to cope with stress.

Materials and methods:

In paragraphs 134-142, the authors should provide more details regarding the experimental conditions (e.g., temperature, dissolved oxygen, pH, water renewal), feeding regime, and tank dimensions.

Regarding both the CuSO4 and trichlorfon exposure experiments, the same details need to be elucidated for the experimental conditions, feeding regime, and tank dimensions. Additionally, provide details regarding the number of administrations of CuSO4 and trichlorfon in the tanks. Was the water being exchanged? Was there any observed mortality during the trials?

Another question that needs clarification is whether both trials were conducted at the same time. Are the fish in group K the same for both experiments? Even if group K is the same for both experiments, the 420 fish from the growth trial are not enough to perform both experiments (at least 450 fish would be needed).

Add a sentence about the analysis of rollover.

Provide more details for the biochemical analysis.

In the section on statistical analysis, were the data checked for ANOVA assumptions? Were the data analyzed using a 2-way ANOVA? Include the broken line analyses.

Results:

In the results section, the authors need to be careful when writing about significant differences regarding the levels of AE. For example, in line 218, it is incorrect to say that H2O2 content was significantly decreased with the increase in dietary AE levels up to 5.0g/kg because the inclusion levels of 1g/kg, 2g/kg, and 3g/kg have no differences in H2O2 content compared to the group K.

In line 207, what does the letter D correspond to in the formula for RWG?

In line 213, remove '(100%)".

In line 227, it should be "4 replicates" instead of "3".

In line 243, correct 'RWG' to 'RFI'.

Discussion:

GPT, LDH, and GOT are biomarkers of cellular necrosis. Usually, an increase in these biomarkers is related to tissue damage. The authors should discuss this relationship.

Only minor suggestions:

Simple summary:

Line 12 - Remove the first 'exposure'.

Line 13 - Replace 'for' with 'in'

Abstract:

Line 23 - Replace 'for' with 'on'.

Reviewer 2 Report

Please see the file.

Minor editing of English language required.

Reviewer 3 Report

1.      The intestine, hepatopancreas and muscle were selected as target tissues respectively in different parts, while the corresponding explanation is absent. In addition, the digestive capability and antioxidant ability were both investigated with their relation or correlation poorly interpreted. This greatly compromise the integrity and systematicness of the present study

2.      The growth performance and feed utilization of fish subjected to the 60-day feeding trial (in the copper and trichlorfon exposure study) should be provided.

3.      For the measurement of digestive enzyme activities: 1) The activity of protease or trypsin should be provided in Table 4 and 6; 2) In Table 6, please justify the detection of digestive enzyme activities in hepatopancreas. Why not conduct this analysis in the intestine?

4.      For the broken-line analysis in Fig. 1-3: 1) The analysis should be re-conducted incorporating the X=0 point (namely the 0.97+0, 0+0.7, and 0+2.2 group in these three sections); 2) Please justify the use of different parameters (RWG, RFI, and IR) in the broken-line analysis in the three parts.

5.      Please justify the use of intestine as the target tissue in the high stocking density stress. Why not conduct the relevant analysis in the hepatopancreas?

6.      The authors stated that “all data were subjected to one-way or two-way ANOVA” (line 186). Please specify which parameters were analyzed by one-way ANOVA, and which were by two-way ANOVA.

7.      Were fish fed during the trichlorfon exposure? If so, please provide the feed consumption data.

8.      The authors stated that “high stocking density may depress digestive ability via disrupting the oxidant-antioxidant status of fish digestive organs” in lines 50-51. More interpretation should be provided in the Introduction or Discussion section.

9.      The discussion section is quite shallow and meanwhile too much descriptive with previous results over-described. The authors should focus mainly on the physiological significance of the changes observed in the analyzed parameters in this study. In addition, the corresponding explanations and/or potential mechanisms should be provided.

10.  The title is too broad to reflect the main contents or results of this study.

11.  Minor ones: 1) Please provide the scientific name of aquatic species when firstly mentioned (for example rainbow trout in line 44). 2) Please state the full name of each abbreviation when they appear for the first time (for example DPPH in line 80).

English in the text is acceptable, but can be polished. 

Round 2

Reviewer 3 Report

     The authors have made substantial efforts to revise the manuscript. All the concerns have been rresponded correctly with the manuscript revised correspondingly. There is only one minor issue: The title should be revised to reveal the three different stressors.